# Community- and trophic-level responses of soil nematodes to removal of a non-native tree at different stages of invasion

**Guadalupe Peralta** [1¤]*, **Ian A. Dickie** [2], **Gregor W. Yeates** [1†], **Duane A. Peltzer** [1]

**1** Manaaki Whenua Landcare Research, Lincoln, New Zealand, **2** Bio-Protection Research Centre, School of Biological Sciences, University of Canterbury, Christchurch, New Zealand

† Deceased.
¤ Current address: School of Biological Sciences, University of Canterbury, Christchurch, New Zealand
* gdlp.peralta@gmail.com

**Data Availability Statement:** Data are available from the Manaaki Whenua data repository at https://doi.org/10.7931/n9cw-z376

## Abstract

Success of invasive non-native plant species management is usually measured as changes in the abundance of the invasive plant species or native plant species following invader management, but more complex trophic responses to invader removal are often ignored or assumed. Moreover, the effects of invader removal at different stages of the invasion process is rarely evaluated, despite a growing recognition that invader impacts are density or stage-dependent. Therefore, the effectiveness of invasive species management for restoring community structure and function across trophic levels remains poorly understood. We determined how soil nematode diversity and community composition respond to removal of the globally invasive tree species *Pinus contorta* at different stages of invasion by reanalysing and expanding an earlier study including uninvaded vegetation (seedlings removed continuously), early invader removal (saplings removed), late removal (trees removed), and no removal (invaded). These treatments allowed us to evaluate the stage-dependent belowground trophic responses to biological invasion and removal. We found that invaded plots had half the nematode taxa richness compared to uninvaded plots, and that tree invasion altered the overall composition of the nematode community. Differences in nematode community composition between uninvaded nematode communities and those under the tree removal strategy tended to dilute higher up the food chain, whereas the composition of uninvaded vs. sapling removal strategies did not differ significantly. Conversely, the composition of invaded compared to uninvaded nematode communities differed across all trophic levels, altering the community structure and function. Specifically, invaded communities were structurally simplified compared to uninvaded communities, and had a higher proportion of short life cycle nematodes, characteristic of disturbed environments. We demonstrate that a shift in management strategies for a globally invasive tree species from removing trees to earlier removal of saplings is needed for maintaining the composition and structure of soil nematode communities to resemble uninvaded conditions.

**Funding:** The New Zealand Ministry of Business, Innovation and Employment provided financial support through the Winning Against Wildings endeavour research programme. The funder had no role in study design, data collection and analysis, decision to publish, or preparation of the manuscript.

**Competing interests:** The authors have declared that no competing interests exist.

## Introduction

Invasive species management programs are widely pursued to prevent or mitigate the impacts that invaders have on native communities (e.g., [1,2]). Despite these efforts, many invasive plant eradication programs either fail to meet their stated management goals or do not rigorously evaluate success (e.g., [3–5]). Those plant invasive removal programs that do measure effectiveness typically focus on responses of the targeted invasive plant species to management *per se* [6], even though assessing the recovery of ecosystems should represent an important part of the evaluation of success, as it is usually an implicit goal of management (e.g., [7]). Some successful invasive eradication programs that have measured ecosystem responses have shown an increase in species richness within a trophic level (e.g., plants) following invasive plant removal [8], whereas others have found that the removal of plant invaders has no net effects, or even negative effects, on native species [9,10].

Because the outcomes of invasive plant species removal, in terms of ecosystem impacts, are variable [11,12], it is essential to evaluate the performance of different management strategies within a system. Previous studies have largely assessed the effectiveness of physical and chemical removal techniques [8,9,12,13], but few have compared removal strategies at different stages of the invasion process, which is needed to determine the best timing for management. Furthermore, most studies that assess the effectiveness of invasive plant eradication programs have considered only the response of plant communities [13–15], with only a few examining changes across trophic levels [6,16,17], even though multitrophic level approaches are necessary to provide management guidance [18]. In addition, invader impacts are usually measured in terms of effects on species numbers [15,19], although more holistic approaches, like the use of ordination techniques, can help to capture community-level effects by quantifying shifts in the composition of species assemblages [20,21].

Among the few studies assessing the effects of invasive plant species removal strategies at different stages of the invasion process, Dickie et al. [16] showed that removing invasive plants at later stages of the invasion process generates a plant community composition that does not resemble communities from either earlier invaded nor uninvaded areas. Conversely, removing saplings allows plant community composition to more closely resemble that of uninvaded areas. Furthermore, Dickie *et al.* demonstrated that different management strategies can cause changes in soil chemistry, soil microbial composition, and the abundance of different invertebrate feeding groups. Despite these significant findings, whether different stage-dependent invasive removal strategies also alter the diversity and composition of soil invertebrate communities remains unknown. Moreover, understanding whether removal strategies at later stages of the invasion process also have a stronger impact on the soil food-web structure and function could be useful for informing management decisions. We therefore, took advantage of the soil biota data collected by Dickie *et al.* [16] to perform an in-depth assessment of soil biota responses to the removal of an invasive plant at different invasion stages.

Within the soil biota, nematodes dominate soil food webs and their functioning [22], providing important information on the structure and complexity of the soil food web. Nematodes are also useful bioindicators of ecosystem processes, resource availability and disturbance of the soil environment [23,24]. Furthermore, because nematodes are intrinsically affected by changes in plant communities [25], they can be greatly affected by plant invasions and plant removal strategies. For example, bacterial feeding nematodes are more abundant in both invaded areas and in areas where invasive trees have been removed, compared to uninvaded areas [16]. Moreover, nematode communities dominated by bacterial feeders are also associated with higher rates of litter decomposition, suggesting that changes in the composition of nematode communities are strongly linked to ecosystem processes such as nutrient

cycling. Despite these changes in the abundance of particular nematode feeding groups [16], it remains unknown how invasive plants affect the diversity, composition, structure and function of nematode communities and whether invasive removal strategies can reverse such impacts. Furthermore, because nematodes are present in different trophic levels, they could be useful indicators of bottom-up cascading effects of invasive plants and invasive removal strategies.

We evaluated the effectiveness of different management strategies of a globally invasive tree species, *Pinus contorta* [26], by using nematode abundance data previously reported in Dickie et al. [16], and additional data on taxa identity and life strategies. We determined whether the diversity and composition of soil nematodes differ amongst invaded (i.e., no removal), early invader removal (i.e., sapling removal), late removal (i.e., tree removal) or uninvaded (i.e., seedling removal) treatments to resolve three hypotheses. (1) Because plant invasions strongly reduce the abundance, and alter the composition of, native plant communities that nematodes largely rely on [14,16], nematode taxa richness will decrease in invaded compared to uninvaded sites, and the nematode community composition will strongly differ between these two treatments. In addition, we expect invaded plots to present a higher proportion of nematode taxa with short life cycles, because these taxa are better adapted to disturbances [27]. (2) After removal of the invasive plant and recolonization by grasses, which represent an important food resource and habitat for nematodes, the richness and composition of the nematode community will recover to resemble uninvaded communities. The removal of the invasive plant should also help to restore the nematode-based soil food web structure and function. Therefore, we expect uninvaded communities, as well as those communities where the invasive plant species was removed, to have a more complex structure, with higher abundance of long life cycle nematode taxa characteristic of undisturbed habitats [27] compared to invaded areas. Furthermore, (3) we expect changes in nematode composition should be observed across all trophic levels of the nematode-based food web due to bottom-up regulation by resource availability [28,29]. Nevertheless, if more advanced stages of the invasion process have stronger legacy effects, the removal of invasive trees might not allow nematode communities to recover and resemble those of uninvaded plots, whereas the removal of saplings would. Finally, we aim to identify which nematode taxa have the greatest contribution to the compositional changes observed, and which removal strategy would be the most appropriate to adopt to restore the soil nematode community after *P. contorta* invasion.

## Methods

Our research was covered under the global concession for research sampling and analysis issued by the New Zealand Department of Conservation to Manaaki Whenua Landcare Research (Permit Number CA-31615-OTH).

### Study site

We studied tree invasions and their management at Craigieburn Forest Park (43˚9'04"S, 171˚ 43'52"E), Canterbury, New Zealand, as described previously in [16]. This area is dominated by southern beech (*Nothofagus solandri var cliffortioides*) and open native shrubland and tussock grassland. Parts of this site were previously used from the 1950s until 1970s as an experimental forest area in which several species of non-native trees were introduced including *Pinus contorta* Loudon (lodgepole pine) [30]. In recent years there have been several different management approaches applied to prevent further spread or local impacts of invasive trees. Within this site, twenty-four 0.04 ha (20 x 20 m) plots with different management histories of the non-native invasive tree species *P. contorta* were selected. From these plots, six had continuous removal of *P. contorta* seedlings (< 2 cm basal diameter) where invasion was prevented by

removing new seedlings at least twice annually ('seedling-removal' plots, i.e., the closest approximation to uninvaded areas); seven plots had *P. contorta* sapling-removal (ca. 10 cm basal diameter and 3–5 m height), which represents the removal of the invader at an early stage ('sapling-removal'); five plots where tree-removal (closed canopy ca. 10 m height stands) was carried out, i.e., invasion was allowed and then later removed ('tree-removal'); and six plots where *P. contorta* trees (closed canopy ca. 10 m height stands) have not been removed ('no-removal', i.e., invaded plots). Removals were done as part of operational management by contractors rather than imposed as an experimental treatment to plot *per se*. Aboveground bio-mass was not removed, which has the advantages of avoiding any export of mineral nutrients as well as being a realistic scenario for either management or natural disturbance, but does result in a temporary increase in organic matter. Furthermore, any attempt at biomass removal would have been inherently incomplete as there is no practical method to remove root bio-mass. Treatments were spatially interspersed and sampling occurred ca. three years after sap-ling- and tree-removal treatments, with some reinvasion occurring at the time of sampling. Reinvasion was most extensive in the sapling removal treatment, with all new invasion < 1 m height and not forming a closed canopy. Additional information about the site and plot selec-tion is provided in Dickie *et al*. [16].

## Sampling

Soil samples were collected and homogenised from five locations within each plot (the centre and at four orthogonal points 7.07 m from the centre) using a 65 mm diameter metal coring device to sample the top 100 mm of mineral soil. Litter was not included in our samples because it was generally sparse, and even when present under the densest pine in the no-removal (invaded) treatment, there was an abrupt transition between litter and mineral soil (i.e., there was no appreciable development of an O soil horizon).

We extracted soil nematodes using the tray method [31] from c. 80 g (dry mass) of each soil sample. We then identified approximately 100 individuals from each sample to nominal genus level and, when further identification was possible, we classified genera into morphospecies. Hereafter we use the term taxa to refer to nematodes ids (i.e., genus and morphospecies). We used the proportion of each taxa as an estimate of the taxa abundance of each sample. After taxonomic identification, we assigned individual nematode taxa to one of six feeding catego-ries according to their feeding habits (plant feeders, plant associated, bacterial feeders, fungal feeders, predators or omnivores) [32]. We designated plant feeder and plant associated nema-todes to the first trophic level of the nematode-based soil food web (TL 1), bacterial and fungal feeders to the second trophic level (TL 2), and predators and omnivores to the third trophic level (TL 3) [33]. Nematodes from TL 2 are microbial feeders and therefore indirectly affected by plant changes, as root exudates influence the microbial community [34]. Similarly, nema-todes from TL 3 can also be indirectly affected by changes in plant composition, as well as directly by changes in nematodes from TL 1 and TL 2, from which they feed.

## Nematode-based ecological indicators

To estimate changes in the structure and function of the nematode-based soil food web, we used three commonly used nematode community level indices. First, we calculated sigma maturity index (ΣMI) [27,35], which is based on the abundance of nematodes' functional guilds [36]. To this end, nematodes were classified according to their life strategies along a coloniser-persister (cp) gradient, where colonisers and persisters are extremes of the cp scale from 1–5 respectively [27]. Briefly, coloniser nematodes are those typically having short life cycles, and under favourable conditions, can rapidly increase in abundance. In contrast,

persister nematodes have long life cycles, low colonisation ability, low reproduction rate, and are more sensitive to habitat disturbances. The ecological indicator ΣMI reflects the proportion of the different cp groups in the community, with higher ΣMI values representing higher proportions of persister nematodes and hence indicating less disturbed environments.

The second ecological indicator we used was the enrichment index (EI), which measures the resource status, i.e., soil fertility, of the ecosystem [23]. Finally, to assess the level of complexity of the community we used the structure index (SI). SI indicates the prevalence of trophic links in the soil food web, where higher SI values indicate higher number of trophic links, i.e., enhanced trophic structure and redundancy [23]. All indices were calculated using the Nematode Indicator Joint Analysis program [37].

## Analyses

To determine whether nematode taxa richness differed across management strategies (seedling-removal, sapling-removal, tree-removal, or no-removal), we used the total number of taxa of the entire community as the response variable in a generalised linear model (GLM). We entered management strategies as a fixed factor (factor with four levels) in the model and used the Poisson error distribution. We also used Tukey's tests to estimate differences between management strategies.

In addition, we tested whether the taxa composition of the entire nematode community differed between management strategies using Permutational Analyses of Variance (PERMANOVA) [38]. To accomplish this, we used two dissimilarity metrics that differ in the emphasis they give to taxa composition vs. relative abundance of each taxon. We used the Jaccard dissimilarity metric, which only uses taxa presence-absence, and the Bray-Curtis dissimilarity metric which also incorporates differences in the relative abundances of taxa. We performed two PERMANOVAs, one with the dissimilarity among plots for the entire nematode community estimated with the Jaccard index as the response variable and management strategies as the predictor. The second PERMANOVA included the dissimilarity among plots for the entire nematode community estimated with the Bray-Curtis index as the response variable, and management strategies as the predictor. We also conducted pairwise multilevel comparisons (with both dissimilarity metrics) to assess differences between individual management strategies. Furthermore, as a way to assess functional changes related to nematodes across management strategies, we compared the nematode-based ecological indicators (ΣMI, EI, SI) across management strategies using ANOVAs and Tukey's tests for comparisons across management strategies.

To evaluate whether different invasive management strategies affected taxa richness and composition of each trophic level (TL), we used GLMs and PERMANOVAs, respectively. Specifically, we performed one GLM (with Poisson error distribution) for each TL, to assess differences in taxa richness across management strategies, and two PERMANOVAs for each TL (one with each dissimilarity metric) to test for differences in taxa composition across management strategies. Finally, we used indicator species analyses to determine which specific nematode taxa mostly drove community composition changes at each trophic level.

All analyses were performed in the R environment [39]. We tested for the overdispersion of residuals assumption of all the Poisson models. We used the 'adonis' and 'betadisper' functions of the vegan package [40] for the PERMANOVAs (9999 permutations), and the 'pairwise.adonis' function [41] for the pairwise multilevel comparisons. We tested for the PERMANOVA homogeneity of multivariate dispersions assumption [42], and used Principal coordinate analysis (PCoA) to illustrate differences between management strategies. We also used the 'multipatt' function from the indicspecies package [43] to perform the indicator species analyses.

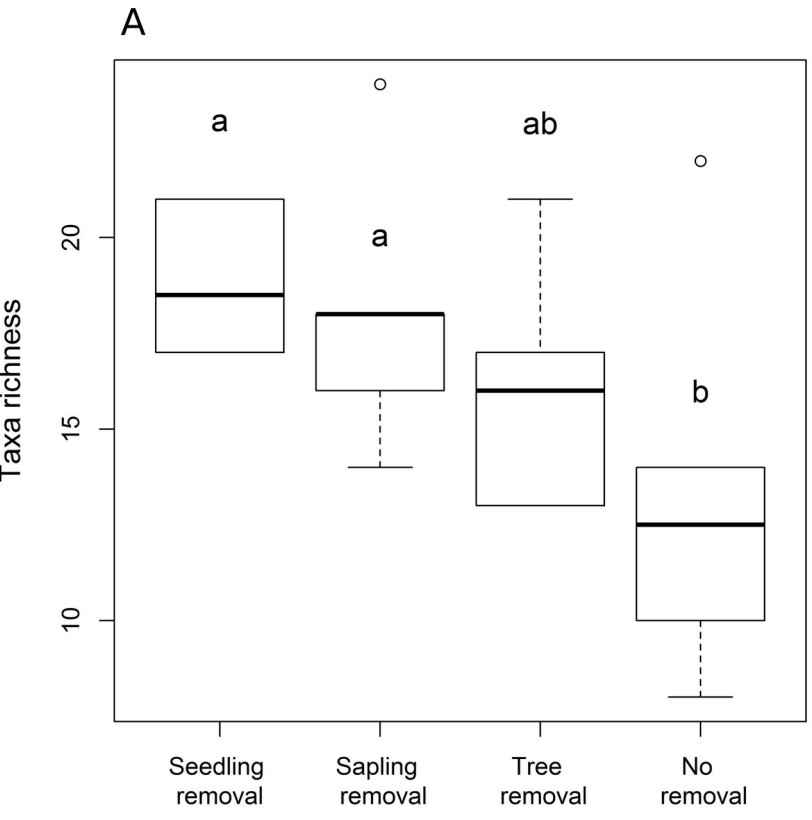

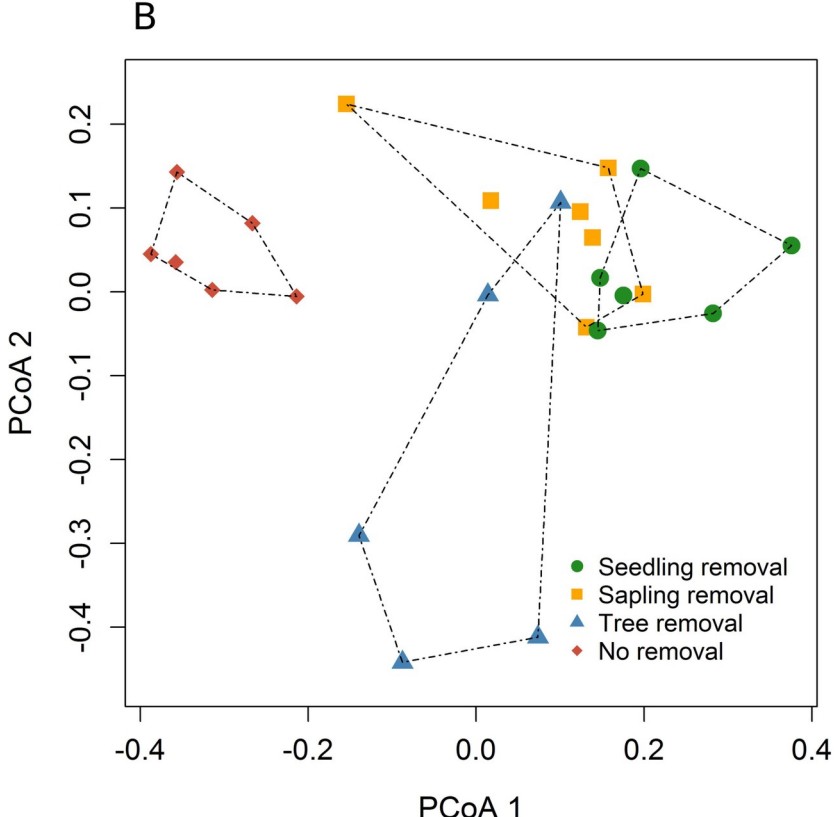

**Fig 1. Responses of total nematode richness and community composition across each of four management strategies for an invasive tree.** Management strategies representing different stages of invasion process: seedling removal, sapling removal, tree removal, no removal. (A) Taxa richness; (B) community composition. (A) Different letters represent significant differences obtained from Tukey's test ($P < 0.05$). In each box plot the middle line indicates the median, bottom and top box limits are the first and third quartiles, respectively, whiskers indicate most extreme points 1.5 times the interquartile range, and circles indicate outliers. (B) Principal Coordinate analyses were based on the Bray-Curtis dissimilarity metric. Sites closer together in multivariate space have similar compositions. Dashed lines represent convex hulls in ordination space.

## Results

Across all management strategies we identified 46 nematode taxa (S1 Table), with an average of $16 \pm 4$ [mean $\pm$ SD] taxa per plot. Across all taxa, 8 belonged to trophic level 1 (TL 1, i.e., plant feeder and plant associated nematodes), 22 to TL 2 (bacterial and fungal feeders) and 16 to TL 3 (predators and omnivores) (S1 Table).

Taxa richness of the overall nematode community was higher in seedling-removal plots only compared to no-removal plots ($Z = -2.441$, $P = 0.015$) (Fig 1A; S2 Table). On the other hand, the overall nematode community composition of seedling-removal plots differed significantly from both no-removal (Pseudo-$F = 9.098$, $P = 0.003$) and tree-removal plots (Pseudo-$F = 2.234$, $P = 0.009$), but not from sapling-removal plots (Pseudo-$F = 0.843$, $P = 0.688$) (Fig 1B; S3 and S4 Tables) when taking into account the relative abundance of taxa (Bray-Curtis dissimilarity). However, when assessing community composition changes by only considering the presence-absence of nematode taxa (Jaccard dissimilarity), only differences between seedling-removal and no-removal plots were observed (Pseudo-$F = 4.147$, $P = 0.002$) (S1 Fig, S3 and S4 Tables).

When testing the effects of invasive removal strategies on the structure and function of the nematode-based food web, we found that two of the three nematode-based ecological indicators assessed differed between the no-removal plots and all other management strategies. More specifically, $\Sigma$MI and structure index (SI) were lower in no-removal plots compared to seedling-, sapling- and tree-removal plots (Fig 2A and 2C; $\Sigma$MI: $F = 22.359$, $P < 0.001$; SI: $F = 21.222$, $P < 0.001$), but no differences were observed for enrichment index (EI) ($F = 1.166$, $P = 0.347$; Fig 2B).

When considering different trophic levels (TLs) separately, we found that taxa richness was at least two-fold higher in seedling-removal and in sapling-removal plots compared to no-removal plots for TL 1 ($Z = -2.404$, $P = 0.016$) (Fig 3A), but did not vary among management strategies for the other trophic levels (Fig 3C and 3E; S2 Table). In addition, the nematode composition of TL 1 in the seedling-removal plots differed significantly from the composition in no-removal plots (Jaccard dissimilarity: Pseudo-$F = 7.487$, $P = 0.006$; Bray-Curtis: Pseudo-$F = 10.123$, $P = 0.002$), but not from the sapling-removal plots (Jaccard dissimilarity: Pseudo-$F = 1.241$, $P = 0.334$; Bray-Curtis: Pseudo-$F = 1.310$, $P = 0.209$) both when using only presence-absence data (Jaccard dissimilarity) as well as when incorporating nematode abundance (Bray-Curtis dissimilarity) (Fig 3B; S3 and S4 Tables). Differences in TL 1 taxa composition between seedling-removal and tree-removal plots were only detected when incorporating the abundance of nematode taxa (Bray-Curtis: Pseudo-$F = 3.330$, $P = 0.006$) (Fig 3B; S4 Table).

For both TL 2 and TL 3, the nematode composition of the seedling-removal plots differed significantly only from that of the no-removal plots, both when considering the presence-absence of nematode taxa (TL 2: Pseudo-$F = 3.006$, $P = 0.007$; TL 3: Pseudo-$F = 3.519$, $P = 0.013$) as well as when incorporating their abundances (TL 2: Pseudo-$F = 9.512$, $P = 0.007$; TL 3: Pseudo-$F = 3.866$, $P = 0.009$) (Figs 3D, 3F, S2B and S2C; S4 Table). The composition of TL 2 from seedling-removal plots also differed from tree-removal plots only when considering

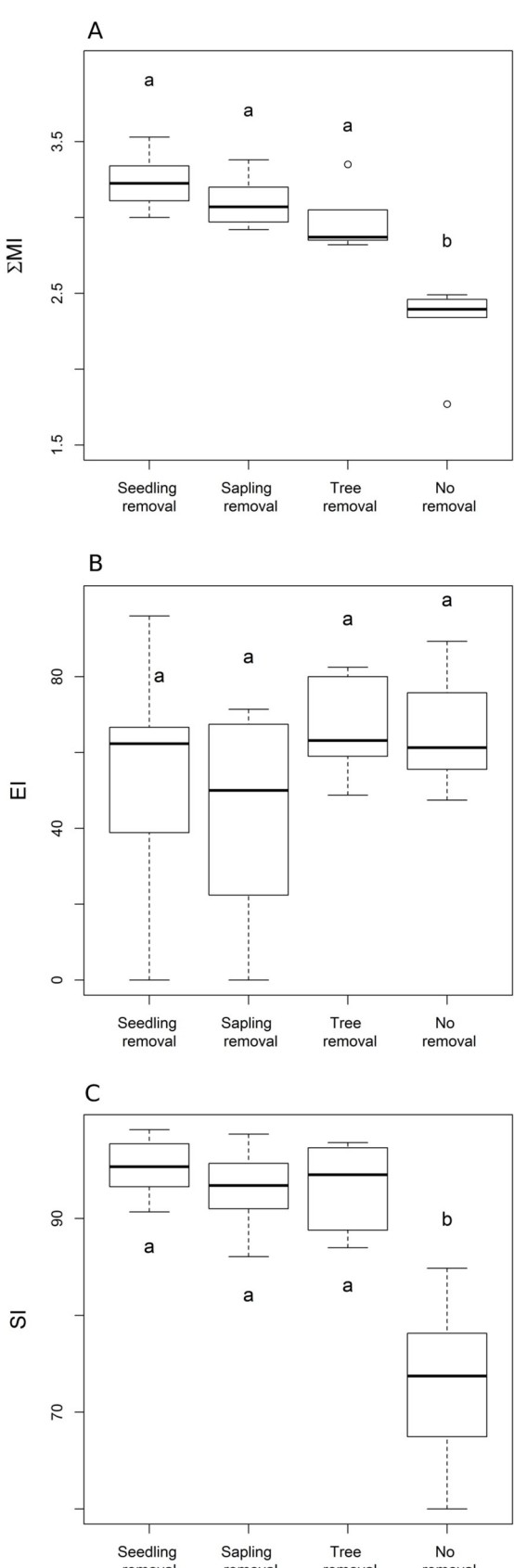

**Fig 2. Nematode community level indices across four management strategies for an invasive tree.** Nematode community indices: (A) ΣMI = sigma maturity index, (B) EI = enrichment index and (C) SI = structure index. Management strategies for an invasive tree representing different stages of the invasion process: seedling removal, sapling removal, tree removal, no removal. Different letters represent significant differences obtained from Tukey's test ($P < 0.05$).

the presence-absence of taxa (Pseudo-$F$ = 2.110, $P$ = 0.048) (S4 Table). Nevertheless, the composition of TL 2 did not differ between seedling-removal plots and sapling-removal plots, both when excluding (Pseudo-$F$ = 0.980, $P$ = 0.466) and including (Pseudo-$F$ = 0.730, $P$ = 0.726) taxa abundance (Figs 3D and S2B; S4 Table).

The indicator species analyses allowed us to identify taxa that were most strongly associated with a management strategy or a group of management strategies (S5 Table). For instance, *Doryllaimellus* (TL 1), *Tylencholaimus* sp 2 (TL 2) and *Aporcelaimidae* (TL 3) were significantly associated with seedling-, sapling- and tree-removal management strategies (Fig 4). In addition, *Doryllium* (TL 2) was a good indicator of seedling-removal plots, whereas *Plectus robus* (TL 2) was strongly associated with sapling-, tree- and no-removal plots (Fig 4).

## Discussion

Multiple management strategies are often deployed in an attempt to reduce the abundance or distribution of invasive species, but with an ultimate goal of avoiding or mitigating negative impacts on native ecosystems (e.g., [44]). Thus, selection of different management strategies should consider both reduction of the invader abundance and the response of native communities [45]. Despite the high relevance of assessing the effects of invasive species removal at different stages in the invasion process, most studies on invasive species impacts consider only the effects of the invader as invasion progresses, but not the effects of invader removal as invasion proceeds [46]. As a contribution to fill this gap in our knowledge, our study demonstrates that both invasion and removal strategies alter the taxonomic composition of the soil nematode community and that invasion also alters community structure and function.

The decrease in the number of nematode taxa with invasion, with lower richness in invaded (no-removal treatment) compared to uninvaded (seedling-removal treatment) plots, is consistent with the decrease in diversity of nematodes and other invertebrates caused by other invasive species [47,48], including other pine species [24], although the opposite pattern has also been observed [49]. In addition, nematode communities from invaded plots differed in taxa composition, suggesting that *P. contorta* exerts a selective pressure over the soil nematode taxa that can inhabit the soil. Furthermore, nematode taxa that were most abundant in invaded plots were those that proliferate under disturbances or stressful conditions, as identified by the decrease in ΣMI. Similar proliferation of nematodes having short life cycles, as well as similar reductions in complexity and redundancy of nematode communities in invaded plots (SI decrease), have also been observed in areas invaded by other tree species [50,51].

An unresolved issue is when invasive species management should be deployed to avoid or reverse negative impacts on communities or ecosystems [20,46]. Our findings demonstrate that relatively early management is needed to avoid impacts of an invasive tree species on the composition of belowground nematode communities. For example, when considering the entire nematode community composition we found that only the sapling-removal strategy resembled uninvaded (seedling removal) plots. Even though sapling removal proved to be a better management strategy for the preservation of the nematode community composition, both sapling- and tree-removal management strategies seem to result in similar community structure. In particular, both removal strategies had similar levels of the structure index (SI) and sigma maturity index (ΣMI) to uninvaded areas, and both indices were higher compared

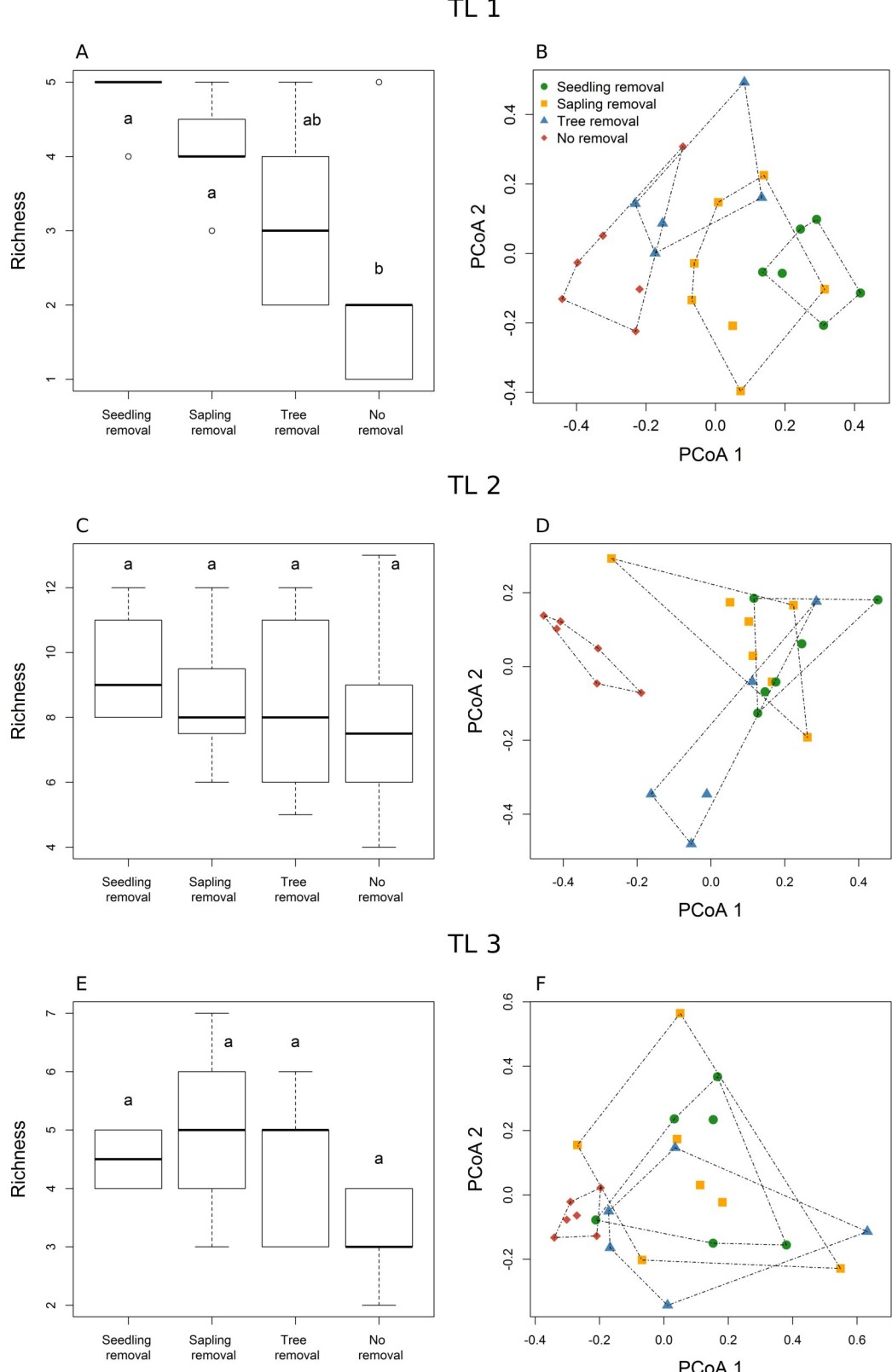

**Fig 3. Nematode taxa richness and community composition of different trophic levels across management strategies.**
Trophic levels: TL1-TL3. Management strategies: seedling removal, sapling removal, tree removal, no removal. Principal

Coordinate analyses were based on the Bray-Curtis dissimilarity metric. Sites closer together in multivariate space have more similar compositions. Different letters in (A), (C), (E) represent significant differences obtained from Tukey's test ($P < 0.05$).

to invaded (no-removal treatment) areas. This suggests that invader removal allows nematode communities to increase their complexity and redundancy to a considerable extent, and also favours nematodes having longer life cycles, which tend to survive only in more stable, less disturbed, environments [23]. Nevertheless, to help restore and preserve the nematode diversity, especially under the tree-removal strategy, further management interventions might be required. To this end, restoration using planting seedlings or increasing the seedbank of native species should be considered [52,53]. Furthermore, inoculation with taxa that were completely eliminated could also be considered [54] to promote restoration of disturbed ecosystems [55]. These interventions could help to restore relatively slow nutrient cycling processes characteristic of uninvaded areas [16,56], and even steer the development of plant communities [55,57].

Changes in the number of taxa is a widely used measure of the impacts and the effectiveness of invasive species removal [9,10,19]. However, we found that the number of taxa (richness) of nematodes failed to capture changes across the upper trophic levels of nematode communities caused by invasion and different removal strategies. Nevertheless, these differences were detected when using multivariate analyses that incorporated only the identity or the identity and abundance of species that are present at a specific place. Differences in nematode community composition among management strategies were strongly influenced by a few taxa. For instance, *Doryllaimellus*, *Tylencholaimus* sp 2 and *Aporcelaimidae* were indicator taxa from seedling- (uninvaded), sapling- and tree-removal management strategies. These three taxa belong to cp group 4 and 5, suggesting that their absence or scarcity in invaded plots is driving the low values of ΣMI in invaded plots. Furthermore, the plant feeder *Doryllaimellus* has also been recorded as absent in areas heavily invaded by *Pinus nigra* [51], suggesting that this taxa could be highly vulnerable to pine invasions in general. In addition, another indicator taxa from uninvaded plots was the fungal feeder *Doryllium*. Uninvaded plots, compared to invaded and afforested areas, tend to have a higher proportion of fungal feeders [24,50,58], which leads to slower decomposition rates compared to bacterial dominated systems [59,60].

Overall, the composition of invaded and uninvaded soil nematode communities differed across all trophic levels (including the highest trophic level, TL 3), suggesting that the impacts of invasive species can cascade up the nematode-based food web. Because our study focuses on a subset of soil organisms, future research should assess the extent to which these findings apply to the entire soil food web, and in turn, feedback to aboveground communities [56].

## Conclusions

Controlling invasive species is complex because it involves a *de facto* manipulation of complex systems [61,62]. However, both the effectiveness of invasive species management and the longer-term recovery of more complex community structure and ecological processes are rarely measured [9]. We demonstrate that a shift in management strategies for a globally invasive tree species (*Pinus contorta*) from removing trees to earlier removal of saplings is needed for maintaining the taxonomic composition of soil nematode communities to resemble uninvaded conditions. In addition, the community-level responses of nematodes to management closely resemble those observed for shifts in plant composition found in the same experiment [16]. Our findings support the early management of invasive species to prevent impacts and

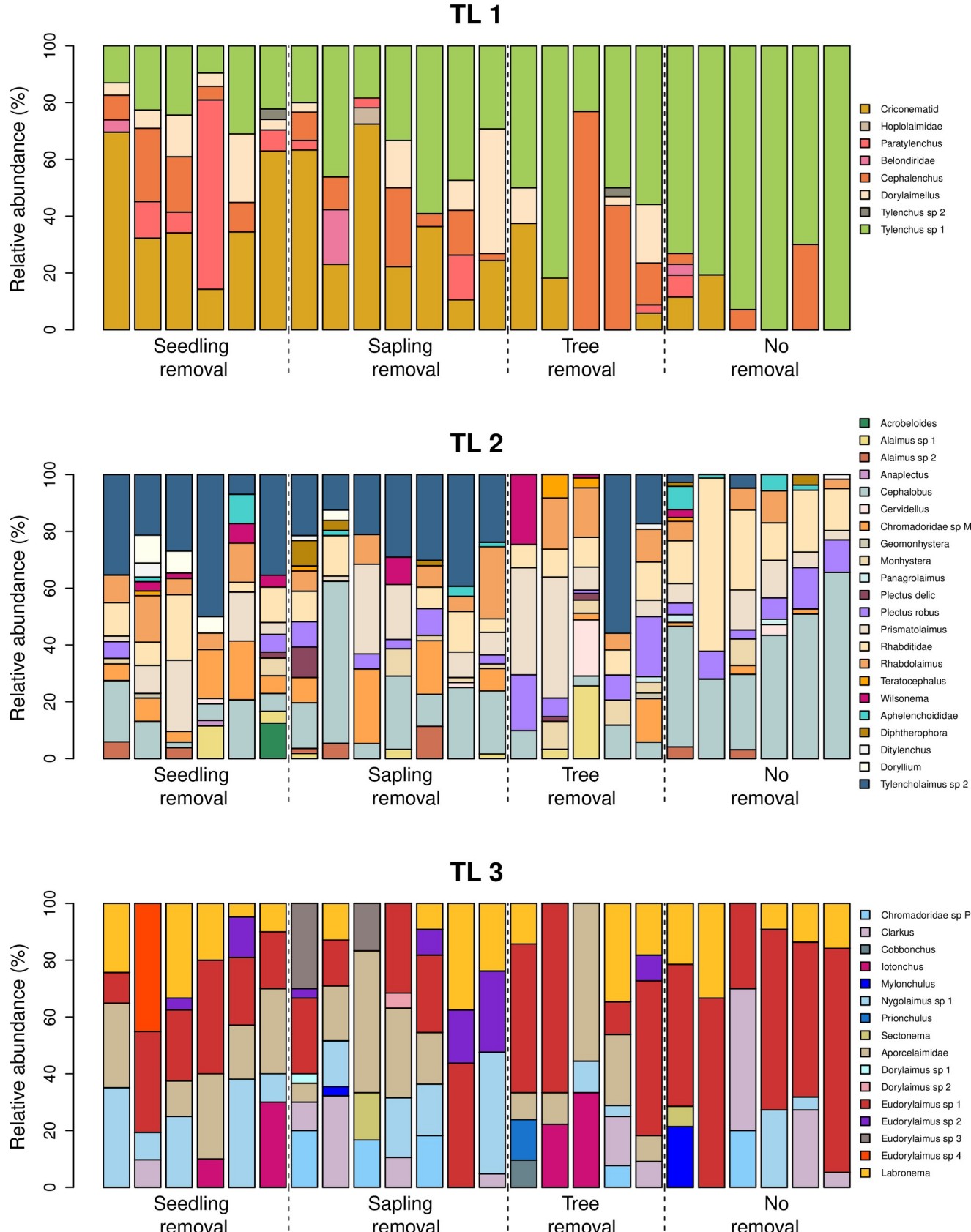

**Fig 4. Relative abundance of nematode taxa (i.e., percentage composition of each nematode taxon within a sampling site) across management strategies and trophic levels.** Trophic levels: TL1-TL3. Management strategies: seedling removal, sapling removal, tree removal, no removal.

potentially strong belowground legacies that can undermine our ability to restore community composition and ecosystem functions over the longer-term.

## Supporting information

**S1 Fig. Nematode community composition across each of four management strategies for an invasive tree.** Management strategies representing different stages of invasion process: seedling removal, sapling removal, tree removal, no removal. Principal Coordinate analyses were based on the Jaccard dissimilarity metric. Sites closer together in multivariate space have similar compositions. Dashed lines represent convex hulls in ordination space.
(PDF)

**S2 Fig. Nematode community composition of different trophic levels across management strategies.** Trophic levels: (A) TL1, (B) TL2, (B) TL3. Management strategies: seedling removal, sapling removal, tree removal, no removal. Principal Coordinate analyses were based on the Jaccard dissimilarity metric. Sites closer together in multivariate space have more similar compositions.
(PDF)

**S1 Table. Nematode classification.** Nematode taxon, feeding group (plant feeder, plant associated, bacterial feeder, fungal feeder, predator, omnivore), trophic level (1–3) to which each taxa was assigned and management strategy where each taxa was present (U = seedling removal, SR = sapling removal, TR = tree removal, NR = No removal).
(DOCX)

**S2 Table. Results of generalised linear models with Poisson error distribution comparing taxa richness of seedling-removal management strategy with taxa richness of sapling-removal, tree-removal and no-removal management strategies.** TL = trophic level. Bold values indicate significant results ($\alpha = 0.05$).
(DOCX)

**S3 Table. Results of PERMANOVA analyses of Jaccard and Bray-Curtis dissimilarity in the entire nematode community and TL 1, TL 2 and TL 3 community composition across different management strategies.** Management strategies: seedling removal, sapling removal, no removal, tree removal. Bold values indicate significant results ($\alpha = 0.05$).
(DOCX)

**S4 Table. Pairwise multilevel comparisons of nematode community composition between different management strategies of the entire nematode community composition and across different trophic levels.** Management strategies: seedling removal, sapling removal, no removal, tree removal. Jaccard and Bray-Curtis dissimilarity metrics were used to assess differences in community composition. Bold values indicate significant differences in community composition ($\alpha = 0.05$).
(DOCX)

**S5 Table. Indicator nematode taxa within trophic level.** Significant indicator taxa of individual or group of management strategies as identified by indicator species analyses for each trophic level. Analyses were performed including taxa abundance and using only presence-absence data. P values were obtained by permuting the data 9999 times. Bold values indicate

significant results ($\alpha = 0.05$).
(DOCX)

## Acknowledgments

We thank C. Morse, R. Buxton, M. St John, H. Maule and K. Boot for substantive field and laboratory assistance; two reviewers for helpful comments; J. Yeates for permission to include Gregor as a co-author on this paper in recognition of his contributions to this work.

## Author Contributions

**Conceptualization:** Ian A. Dickie, Gregor W. Yeates, Duane A. Peltzer.

**Data curation:** Ian A. Dickie, Gregor W. Yeates, Duane A. Peltzer.

**Formal analysis:** Guadalupe Peralta.

**Methodology:** Ian A. Dickie, Gregor W. Yeates, Duane A. Peltzer.

**Writing – original draft:** Guadalupe Peralta.

**Writing – review & editing:** Ian A. Dickie, Duane A. Peltzer.

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
