## [Decision Letter · Decision Letter 0]

20 Aug 2019

PONE-D-19-17858

Multi-trophic level responses of belowground nematode communities to stage dependent removal strategies of an invasive tree

PLOS ONE

Dear Miss Peralta,

Thank you for submitting your manuscript to PLOS ONE. After careful consideration, we feel that it has merit but does not fully meet PLOS ONE’s publication criteria as it currently stands. Therefore, we invite you to submit a revised version of the manuscript that addresses the points raised during the review process.

We would appreciate receiving your revised manuscript by Oct 04 2019 11:59PM. To enhance the reproducibility of your results, we recommend that if applicable you deposit your laboratory protocols in protocols.io, where a protocol can be assigned its own identifier (DOI) such that it can be cited independently in the future. For instructions see: http://journals.plos.org/plosone/s/submission-guidelines#loc-laboratory-protocols

We look forward to receiving your revised manuscript.

Kind regards,

Martin Schädler

Academic Editor

PLOS ONE

Journal Requirements:

3. We noted in your submission details that a portion of your manuscript may have been presented or published elsewhere: "Part of the data has been used in a previous publication, as explained in the cover letter and in the manuscript (introduction and material and methods)."

Additional Editor Comments:

Both reviewers are pretty positive regarding the study itself and the results but both recommend substantial text work in more or less all parts of the manuscript.

Reviewers' comments:

Reviewer's Responses to Questions

**Comments to the Author**

1. Is the manuscript technically sound, and do the data support the conclusions?

Reviewer #1: Partly

Reviewer #2: Yes

2. Has the statistical analysis been performed appropriately and rigorously? 

Reviewer #1: No

Reviewer #2: Yes

3. Have the authors made all data underlying the findings in their manuscript fully available?

Reviewer #1: No

Reviewer #2: No

4. Is the manuscript presented in an intelligible fashion and written in standard English?

Reviewer #1: Yes

Reviewer #2: No

5. Review Comments to the Author

Reviewer #1: The reviewed manuscript describes the results of an experiment examining the responses of soil nematode communities to an invasive plant (Pinus contorta) management intensity gradient. I am always excited to see new studies examining the responses of soil nematodes to plant invasion and management. The results presented here are interesting and highlight the importance of considering soil community responses to plant invasion and management. This manuscript presents data from an experiment originally described in Dickie et al. 2014 (AoB Plants). I remain at least marginally concerned about potential overlap between this MS and Dickie at al. 2014, which also presents some soil nematode data. Dickie et al. 2014 reports abundance data of nematode feeding guilds (bacterial-feeding, fungal-feeding, etc.), but community composition and diversity/richness data were not reported. Thus in my opinion, the data presented here are a substantial expansion of the Dickie et al. 2014 paper, though not entirely independent.

That being said, I would recommend restructuring the Introduction to better frame this work as a direct expansion of Dickie et al. 2014. That paper is cited six times prior to the reveal that this MS is an in-depth assessment of nematodes initially reported in that paper. Over that same stretch of MS, no other paper is cited more than twice. So this MS relies heavily on Dickie et al. 2014. For the sake of transparency, I’d recommend acknowledging that upfront. Additionally, maintaining similar terminology between Dickie et al. 2014 and this MS would facilitate an easier comparison between the two. For example, Dickie et al. 2014 refers to the four treatments as 1) seedling-removal, 2) sapling-removal, 3) no-removal, and 4) tree-removal. In this MS, the four groups are 1) uninvaded, 2) sapling-removal, 3) tree-removal, and 4) invaded.

Additional methodological details are needed to fully assess experimental rigor. I also recommend new analyses to provide insights into nematode community responses to invasion and management.

Abstract:

-No direction or magnitude of invasion or management effects are included in the abstract. I recommend adding statements along the lines of “Invaded plots had XX% lower richness than seedling removal plots.”

Introduction:

-The hypotheses focus primarily on nematode richness and community composition. Composition data typically includes taxon abundance, which may be encroaching on the abundance data presented in Dickie et al. 2014.

Methods:

-The phrasing ‘we selected‘ plots (line 118) sounds strange. Were these not the same 24 plots established in Dickie et al. 2014?

-Is removal of seedlings twice a year often enough to be considered ‘uninvaded’? I’d like to see more data on the number of seedlings removed. Early removal of 3 vs. 3000 seedlings would be an important factor in determining if you are maintaining this area as uninvaded, or if it is well into the early stages of invasion.

-Why no removal of invasive plant biomass from the site (line 127)? I need some justification for this. With herbaceous plant biomass, it is more typical to remove the biomass since its decomposition will alter soil nutrients and soil fauna.

-Sampling occurred three years after removal treatments, but more details are needed. Was sampling three years after the initiation of removal treatments if removal treatments were a one-time event? Seedling removal seems like it would need to be recurring regularly, which it may have been (line 121), but I’m still not clear on this. If reinvasion as occurred (lines 128-129) has this shifted plots from one group to another?

-I have some concerns about the designation of trophic groups T1, T2, and T3. As presented, it invokes a linear food chain with T1 being directly associated with the plant (by feeding on the plant) with T2 and T3 being sequentially further removed from direct plant interactions. In reality, these are two pathways of energy transfer from the plant, either to 1) plant-feeding nematodes directly, or to 2) microbial-feeding nematodes indirectly via root exudation and root turnover stimulating bacteria and fungi, which are then consumed by nematodes. Predatory nematodes may serve to bridge these two energy pathways by feeding on plant-feeding, bacterial-feeding, and fungal-feeding nematodes. I’d argue that the term ‘trophic level’ is not appropriate, and perhaps use ‘trophic group’.

-Were nematode analyses run on genera or morpho-species data? Line 172 states that number of species were used, but nematodes were only identified to genus or morpho-species (lines 144-146).

-It seems that relative abundance data was used for the ordinations and PERMANOVA (line 181). Is this correct? I highly recommend use of absolute abundance data, since differences can be caused by both changes in species identity but also abundance, though use of abundance data might overlap more with the Dickie et al. 2014 paper.

-Following up on my earlier comment regarding the designated trophic levels, I think the SEM needs justification for including a direct effect of plants on TL3. In a soil food web, effects of plants on predaceous nematodes would likely be modulated by the lower trophic levels (TL1 and TL2). Direct effects of management could still exist on TL3, likely through habitat disruption, and could be included in the SEM.

Results:

-My biggest concern with the Results is that no data are presented to show which nematode species, or trophic groups, are driving the observed changes in composition. The PERMANOVA is great for testing for differences between groups, and the PCoA is one method for visualizing differences among groups, but for food webs it is important to know which genera/trophic groups are driving the observed changes. Without these data, we have an incomplete picture of the soil nematode responses to invasion and management. Ideally, inclusion of additional analyses, such as an indicator species analysis, or at minimum, a breakdown of feeding groups and/or trophic levels by treatment are needed.

Discussion:

-The Discussion generally lacks comparison of these results to previously published studies examining soil nematode responses to invasive plant management (lines 289-308).

-Are there comparisons you can make with other published studies examining the timing of invasive plant management on other organisms? Comparisons to aboveground fauna may also be of interest to assess the generality of invasion and management effects in higher trophic levels.

Data accessibility:

-No database is listed where data will be deposited upon successful publication.

Minor comments:

-line 253: remove extra period after Invaded

-line 304: change ‘same area’ to ‘same experiment’

-line 330: change ‘than’ to ‘to’

Reviewer #2: 1. The manuscript reports changes of soil nematode community structure in response to ontogenic removal of an invasive tree species. The study was well designed and the methodology was proper to address the concerns. However, this manuscript is overall poorly written and may need reorganization.

2. The title is confusing (e.g. stage-dependent, what stages?) and not informative enough as the authors estimated the responses of soil nematodes both at community (composition, indices etc.) and trophic level to the management strategies. I suggest it could be changed like “Community- and trophic-level responses of soil nematodes to removal of an invasive tree species at different invasion stages”.

3. Sap- and tree-removal are the two comparable management strategies (treatments) employed in this study. However, in the introduction I did not read any information about “why the two strategies were comparably tested” or “what are the potential difference in responses of soil food webs to sap- and tree-removal”.

4. The hypotheses need to be adjusted: (1) Line 92-93: I could not read how the nematode species richness will increase if invasions reduce the abundance and alter the composition of native plant communities. I guess the authors attempted to highlight the invasions may reduce the abundance and diversity of native plant communities, right? (2) I understand that removal of invasive trees would results in the recolonization of herbs, which may help restore the nematode communities…However, I am confused by the following sentence “We expect uninvaded communities to have more complex structure, with higher abundance of long-life cycle nematode taxa…”. Does that mean uninvaded habitat will have more complex structure or higher abundance of nematodes than “invaded habitat”?

5. The authors claimed that the removal management strategies influence the soil nematodes via effects on plant community, such as indicated in Fig. 4, but I could not find any data or related analysis on plant community. I strongly suggest them to include data on plant community composition they scored (percentage of coverage). Otherwise, I could not see the value of Fig. 4 in this study.

6. I would change the structure of the manuscript, showing the community- and trophic- level responses separately. Fig. 1a, b and Fig. 3 showed tree species invasion indeed resulted in a different nematode community responses from uninvaded soil, and both sap- and tree-removal could reverse this disturbance of invasion to the level of uninvaded treatment. On the other hand, at the trophic level, it seems only sap-removal of the invasive tree species could reverse the disturbance (represented as nematode community responses) back to the level of the uninvaded treatment (Fig. 1c and Fig. 2), suggesting different trophic levels would respond to management strategies in different ways.

7. I found the results were not fully discussed in the current version. I would like to read discussion on (1) invaded vs. uninvaded: tree invasion indeed alter soil nematode communities; (2) removal of these invasive trees could reverse these invasion influences; (3) the extent of removal of invasive trees recovering nematode communities depends on the invasion stage of removal.

8. Specific comments:

L62: what functional traits?

L73: NOT a sentence

L83-86: not logic

L90: “changes” should be “differs”

L100-102: confusing. What is the comparison?

L128-129: what do you want to say by mentioning “with some reinvasion…”?

L143: where is the data of soil moisture?

L167: more details are needed on SI to introduce this index.

L172: “total number of species”? I thought authors mostly identify the nematodes at genus level

L319-321: not understandable

L329: “have” should be “result in”

L331: “both” should be “all three”

L334: “perturbations” should be “perturbations, such as plant invasions”

L340: should be “soil nematode communities” instead of “communities”

6. PLOS authors have the option to publish the peer review history of their article (what does this mean?). If published, this will include your full peer review and any attached files.

Reviewer #1: No

Reviewer #2: No

---

## [Author Response · Author response to Decision Letter 0]

10 Oct 2019

>> Dear Associate Editor and reviewers: the line numbers in our responses refer to the line numbers of the manuscript with tracked changes.

Journal Requirements:

http://www.journals.plos.org/ and http://www.journals.plos.org/

>> Our manuscript meets the PLOS ONE's style requirements.

>> Upon manuscript acceptance we will provide the data in an on-line repository (the datastore system hosted by Manaaki Whenua, https://datastore.landcareresearch.co.nz) to facilitate access and use of this information.

3. We noted in your submission details that a portion of your manuscript may have been presented or published elsewhere: "Part of the data has been used in a previous publication, as explained in the cover letter and in the manuscript (introduction and material and methods)."

>> As we explain in the manuscript and in the first cover letter (i.e. when we submitted the manuscript for the first time), we use nematode data that was included in a very broadly focused paper (Dickie et al. 2014, AoB Plants). In this previous publication, only abundances of different nematode trophic groups were published; most results in that paper focused on shifts in plant community composition and soil nutrient availability. In our study we provide results for more detailed analyses that include additional information about nematode species identities and life strategies; this allows us, for the first time, to estimate the overall nematode-based food-web composition and structure. Thus, while some of the underlying data was included in a previous paper, this new manuscript presents (a) new data on species identity and life strategies, (b) new theoretical and analytical approaches to the data, and (c) an entirely new understanding of nematode community composition following invasion.

Additional Editor Comments:

Both reviewers are pretty positive regarding the study itself and the results but both recommend substantial text work in more or less all parts of the manuscript.

>> We appreciate the positive feedback. We have incorporated the comments made by the reviewers and responded specifically to each of them below, marking our responses with ‘>>’.

Reviewers' comments:

Reviewer's Responses to Questions

Comments to the Author

1. Is the manuscript technically sound, and do the data support the conclusions?

Reviewer #1: Partly

Reviewer #2: Yes

>> We have clarified all the points made by reviewer #1 in the methods section.

2. Has the statistical analysis been performed appropriately and rigorously?

Reviewer #1: No

Reviewer #2: Yes

>> We have now incorporated the additional analyses suggested by reviewer #1.

3. Have the authors made all data underlying the findings in their manuscript fully available?

Reviewer #1: No

Reviewer #2: No

>> We will provide the data in an on-line repository (the datastore system hosted by Manaaki Whenua, https://datastore.landcareresearch.co.nz) to facilitate access and use of this information upon manuscript acceptance.

4. Is the manuscript presented in an intelligible fashion and written in standard English?

Reviewer #1: Yes

Reviewer #2: No

>> We have re-written several sections of the manuscript and re-structured the discussion, as suggested by reviewer #2.

5. Review Comments to the Author

Reviewer #1: The reviewed manuscript describes the results of an experiment examining the responses of soil nematode communities to an invasive plant (Pinus contorta) management intensity gradient. I am always excited to see new studies examining the responses of soil nematodes to plant invasion and management. The results presented here are interesting and highlight the importance of considering soil community responses to plant invasion and management. This manuscript presents data from an experiment originally described in Dickie et al. 2014 (AoB Plants). I remain at least marginally concerned about potential overlap between this MS and Dickie at al. 2014, which also presents some soil nematode data. Dickie et al. 2014 reports abundance data of nematode feeding guilds (bacterial-feeding, fungal-feeding, etc.), but community composition and diversity/richness data were not reported. Thus in my opinion, the data presented here are a substantial expansion of the Dickie et al. 2014 paper, though not entirely independent.

That being said, I would recommend restructuring the Introduction to better frame this work as a direct expansion of Dickie et al. 2014. That paper is cited six times prior to the reveal that this MS is an in-depth assessment of nematodes initially reported in that paper. Over that same stretch of MS, no other paper is cited more than twice. So this MS relies heavily on Dickie et al. 2014. For the sake of transparency, I’d recommend acknowledging that upfront.

Additionally, maintaining similar terminology between Dickie et al. 2014 and this MS would facilitate an easier comparison between the two. For example, Dickie et al. 2014 refers to the four treatments as 1) seedling-removal, 2) sapling-removal, 3) no-removal, and 4) tree-removal. In this MS, the four groups are 1) uninvaded, 2) sapling-removal, 3) tree-removal, and 4) invaded.

>> As suggested by the reviewer, we now acknowledge upfront that this study represents an expansion of Dickie et al. 2014, in which we provide an in-depth assessment of soil biota responses to different removal strategies of a widespread invasive tree species (L28, 72-86). In addition, we have changed the terminology in our manuscript to make it consistent with that of Dickie et al. 2014, to facilitate comparison between these studies.

Additional methodological details are needed to fully assess experimental rigor. I also recommend new analyses to provide insights into nematode community responses to invasion and management.

>> We have modified the methods section according to the reviewer’s suggestions and also incorporated the new analysis suggested by the reviewer (also see below for additional details).

Abstract:

-No direction or magnitude of invasion or management effects are included in the abstract. I recommend adding statements along the lines of “Invaded plots had XX% lower richness than seedling removal plots.”

>> We have incorporated more specific results in the abstract as suggested by the reviewer (L32-33, 36-37, 39-41).

Introduction:

-The hypotheses focus primarily on nematode richness and community composition. Composition data typically includes taxon abundance, which may be encroaching on the abundance data presented in Dickie et al. 2014.

>> We have rephrased the hypotheses following the suggestions of reviewer #2. We have also incorporated the idea of finding out which nematode taxa are mostly driving the community composition changes, as suggested by reviewer #1. Furthermore, we have now incorporated a community composition analysis using the Jaccard dissimilarity index, which basically evaluates differences in the composition of communities by using only presence-absence data. With this new analysis we show that changes in nematode community composition between invaded and uninvaded plots are also evident when only taking into account the identity (i.e. presence-absence) of taxa (L291-294, 331-337, 340-343, 345-353). 

 Although we acknowledge that we use abundance data that was also used in Dickie et al. 2014, we use abundance data at the taxa (genus or morpho-species) level, whereas Dickie et al. (2014) evaluated changes in the abundance of nematode feeding groups (i.e. not of each taxa independently, but at the feeding group level). Finally, to avoid any potential overlap with data reported by Dickie et al. we have now removed the trophic group composition analysis (previous Fig. 1c).

Methods:

-The phrasing ‘we selected‘ plots (line 118) sounds strange. Were these not the same 24 plots established in Dickie et al. 2014?

>> We have rephrased this sentence.

-Is removal of seedlings twice a year often enough to be considered ‘uninvaded’? I’d like to see more data on the number of seedlings removed. Early removal of 3 vs. 3000 seedlings would be an important factor in determining if you are maintaining this area as uninvaded, or if it is well into the early stages of invasion.

>>This is a good point. We did not carry out the seedling removal treatment ourselves, but rather this has been carried out by volunteer groups who did not record data for number of seedlings removed during each management event. We have assessed seedling densities in the ‘uninvaded’ area, and typically observe 1-2 seedlings at a 20 x 20 m plot scale (or 25-50 seedlings per ha). We have clarified the ‘uninvaded’ treatment in the text to explain this is not zero or never-invaded, but is relatively uninvaded compared to the other management treatments we considered (160-161). Moreover, we also refer to Dickie et al. 2011 and Peralta et al. 2019 where such low densities of tree invasion have minimal or no detectable impacts on belowground communities.

-Why no removal of invasive plant biomass from the site (line 127)? I need some justification for this. With herbaceous plant biomass, it is more typical to remove the biomass since its decomposition will alter soil nutrients and soil fauna.

>> The reviewer raises another valid point. Removals were done as part of operational management at the site rather than imposed strictly as an experimental treatment. Because it is not practical or feasible for managers to remove (aboveground) tree biomass from a controlled site, the normal management practice is to leave biomass in situ. This practice has the advantages of avoiding any export of mineral nutrients as well as being a realistic scenario for either management or natural disturbance, although it does result in a temporary increase in organic matter. In addition, any attempt at biomass removal would have been inherently incomplete as there is no practical method to remove root biomass. We now state this clearly in the revised methods section (L167-173).

-Sampling occurred three years after removal treatments, but more details are needed. Was sampling three years after the initiation of removal treatments if removal treatments were a one-time event? Seedling removal seems like it would need to be recurring regularly, which it may have been (line 121), but I’m still not clear on this. If reinvasion as occurred (lines 128-129) has this shifted plots from one group to another?

>> As mentioned above, removal treatments were imposed by weed managers at the site. The tree removals were done within a single season by contractors, whereas the seedling removal treatment for the ‘univaded’ treatment was carried out by a volunteer group twice per year and is still ongoing. Because we took advantage of these operational management practices that had already occurred, we could not control the fact that ca. three years have passed since the sapling- and tree-removal treatments. Although there was some reinvasion at the time of sampling (L173-177), all new invasion was < 1 m height and not forming a closed canopy.

-I have some concerns about the designation of trophic groups T1, T2, and T3. As presented, it invokes a linear food chain with T1 being directly associated with the plant (by feeding on the plant) with T2 and T3 being sequentially further removed from direct plant interactions. In reality, these are two pathways of energy transfer from the plant, either to 1) plant-feeding nematodes directly, or to 2) microbial-feeding nematodes indirectly via root exudation and root turnover stimulating bacteria and fungi, which are then consumed by nematodes. Predatory nematodes may serve to bridge these two energy pathways by feeding on plant-feeding, bacterial-feeding, and fungal-feeding nematodes. I’d argue that the term ‘trophic level’ is not appropriate, and perhaps use ‘trophic group’.

>> Although the definition of trophic group is ‘a group of organisms consuming resources from a similar level in the energy cycle’, and hence the reviewer is correct that could apply to our classification, in nematode studies the term trophic group is usually associated to feeding types (i.e. plant-feeders, bacterial-feeders, etc.). Because in our classification we are combining different feeding types, we prefer to keep the term trophic level, to avoid any potential confusion with specific feeding types. The assignment of different nematode feeding groups to different trophic levels is well spread, from institutions such as the United States Department of Agriculture (https://www.nrcs.usda.gov/wps/portal/nrcs/detailfull/soils/health/biology/?cid=nrcs142p2_053866), to scientific nematode studies (e.g. Laliberte et al. Ecology Letters 2017). Nevertheless, we acknowledge that nematodes from higher trophic levels also depend indirectly on plants and, therefore, we have clarified this point in our manuscript (L200-203).

-Were nematode analyses run on genera or morpho-species data? Line 172 states that number of species were used, but nematodes were only identified to genus or morpho-species (lines 144-146).

>> Nematode analyses were run using the maximum classification possible for each taxa, i.e. genus or morphospecies. We recognised the term species was not accurate in line 172 and have now replaced it by the term taxa. In addition, we have clarified in the methods (previous lines 144-146) that we used the term taxa to refer to the maximum classification possible (L193-194).

-It seems that relative abundance data was used for the ordinations and PERMANOVA (line 181). Is this correct? I highly recommend use of absolute abundance data, since differences can be caused by both changes in species identity but also abundance, though use of abundance data might overlap more with the Dickie et al. 2014 paper.

>> Because the Bray-Curtis dissimilarity index is calculated based on the relative abundance of the different taxa, the use of absolute abundance data would reflect the same differences among treatments. In addition, to assess whether the differences in nematode community composition across treatments were mostly driven by changes in the relative abundance of taxa or whether the presence-absence of taxa was also a significantly important factor, we have now incorporated additional analyses using the Jaccard dissimilarity index. This index estimates the dissimilarity between communities based on the presence-absence of taxa only, disregarding the taxa relative abundance which the Bray-Curtis index incorporates. By using both of these indices, we show that not only changes in the relative abundance of taxa, but also changes in the identities of the nematode taxa are driving differences in community composition between treatments.

-Following up on my earlier comment regarding the designated trophic levels, I think the SEM needs justification for including a direct effect of plants on TL3. In a soil food web, effects of plants on predaceous nematodes would likely be modulated by the lower trophic levels (TL1 and TL2). Direct effects of management could still exist on TL3, likely through habitat disruption, and could be included in the SEM.

>> Because additional analyses on plant species composition cannot be added to this manuscript, as suggested by reviewer #2, as these are already published (Dickie et al. 2014), we decided to remove this analysis. We believe removing this analysis does not affect our conclusions and it also allows us to incorporate the additional species indicator analysis suggested by the reviewer.

Results:

-My biggest concern with the Results is that no data are presented to show which nematode species, or trophic groups, are driving the observed changes in composition. The PERMANOVA is great for testing for differences between groups, and the PCoA is one method for visualizing differences among groups, but for food webs it is important to know which genera/trophic groups are driving the observed changes. Without these data, we have an incomplete picture of the soil nematode responses to invasion and management. Ideally, inclusion of additional analyses, such as an indicator species analysis, or at minimum, a breakdown of feeding groups and/or trophic levels by treatment are needed.

>> We have now included indicator species analyses (L255-256), as suggested by the reviewer. We believe with these additional analyses and the new Figure 4, it becomes clear which are the species that have the highest influence in the differences between community compositions across treatments.

Discussion:

-The Discussion generally lacks comparison of these results to previously published studies examining soil nematode responses to invasive plant management (lines 289-308).

>> We now compare our results with those of previous studies examining soil nematode responses to plant invasions and management (e.g. Meisner et al. 2014, Čerevková et al. 2019, Peralta et al. 2019).

-Are there comparisons you can make with other published studies examining the timing of invasive plant management on other organisms? Comparisons to aboveground fauna may also be of interest to assess the generality of invasion and management effects in higher trophic levels.

>> Despite the huge literature available on biological invasions and their ecology or management, there are very few that consider the effects of removal at different stages in the invasion process (see review by Simberloff et al. 2013). Rather, most studies on impacts consider only the effects of the invader as invasion progresses and the abundance or biomass of the invader increases, but not the effects of invader removal or management as invasion proceeds. We now mention this in the revised discussion (L406-409).

Data accessibility:

-No database is listed where data will be deposited upon successful publication.

>> As we mentioned previously, we will provide the data in an on-line repository (the datastore system hosted by Manaaki Whenua, https://datastore.landcareresearch.co.nz) to facilitate access and use of this information.

Minor comments:

-line 253: remove extra period after Invaded

>> Done.

-line 304: change ‘same area’ to ‘same experiment’

>> Done.

-line 330: change ‘than’ to ‘to’

>> Done.

Reviewer #2: 1. The manuscript reports changes of soil nematode community structure in response to ontogenic removal of an invasive tree species. The study was well designed and the methodology was proper to address the concerns. However, this manuscript is overall poorly written and may need reorganization.

>> We have re-written several parts of the manuscript, clarified our hypotheses and re-structured the discussion as suggested by the reviewers.

2. The title is confusing (e.g. stage-dependent, what stages?) and not informative enough as the authors estimated the responses of soil nematodes both at community (composition, indices etc.) and trophic level to the management strategies. I suggest it could be changed like “Community- and trophic-level responses of soil nematodes to removal of an invasive tree species at different invasion stages”.

>> We have changed the title as suggested by the reviewer.

3. Sap- and tree-removal are the two comparable management strategies (treatments) employed in this study. However, in the introduction I did not read any information about “why the two strategies were comparably tested” or “what are the potential difference in responses of soil food webs to sap- and tree-removal”.

>> We have now incorporated in the hypotheses the potential difference in responses of soil food webs to sap- and tree-removal treatments (L137-139).

4. The hypotheses need to be adjusted: (1) Line 92-93: I could not read how the nematode species richness will increase if invasions reduce the abundance and alter the composition of native plant communities. I guess the authors attempted to highlight the invasions may reduce the abundance and diversity of native plant communities, right? (2) I understand that removal of invasive trees would results in the recolonization of herbs, which may help restore the nematode communities…However, I am confused by the following sentence “We expect uninvaded communities to have more complex structure, with higher abundance of long-life cycle nematode taxa…”. Does that mean uninvaded habitat will have more complex structure or higher abundance of nematodes than “invaded habitat”?

>> We apologise for the lack of clarity of our hypotheses. We have now rephrased them to improve their comprehensibility (L116-143).

5. The authors claimed that the removal management strategies influence the soil nematodes via effects on plant community, such as indicated in Fig. 4, but I could not find any data or related analysis on plant community. I strongly suggest them to include data on plant community composition they scored (percentage of coverage). Otherwise, I could not see the value of Fig. 4 in this study.

>> We agree with the reviewer that the presence of the plant community variable in the previous Fig 4 does not make sense if we do not present plant community data in other forms in the manuscript. Given that the plant community data has already been published in a previous publication (Dickie et al. AoB Plants, 2014), we decided to remove this analysis and instead focus on the additional analyses suggested by reviewer #1.

6. I would change the structure of the manuscript, showing the community- and trophic- level responses separately. Fig. 1a, b and Fig. 3 showed tree species invasion indeed resulted in a different nematode community responses from uninvaded soil, and both sap- and tree-removal could reverse this disturbance of invasion to the level of uninvaded treatment. On the other hand, at the trophic level, it seems only sap-removal of the invasive tree species could reverse the disturbance (represented as nematode community responses) back to the level of the uninvaded treatment (Fig. 1c and Fig. 2), suggesting different trophic levels would respond to management strategies in different ways.

>> We have re-structured the manuscript as suggested by the reviewer. Specifically, we now present first the results on composition, structure and function of the entire nematode communities under different treatments, and lastly the changes observed in the composition of the different trophic levels.

7. I found the results were not fully discussed in the current version. I would like to read discussion on (1) invaded vs. uninvaded: tree invasion indeed alter soil nematode communities; (2) removal of these invasive trees could reverse these invasion influences; (3) the extent of removal of invasive trees recovering nematode communities depends on the invasion stage of removal.

>> We have re-structured the discussion as suggested by the reviewer.

8. Specific comments:

L62: what functional traits?

>> We have removed this sentence from the introduction.

L73: NOT a sentence

>> We have rephrased this sentence.

L83-86: not logic

>> We have revised the text.

L90: “changes” should be “differs”

>> We have changed this as suggested by the reviewer.

L100-102: confusing. What is the comparison?

>> We have now rephrased this sentence.

L128-129: what do you want to say by mentioning “with some reinvasion…”?

>> A few (literally) newly emerged seedlings had established within the 20 x 20 m area sampled. We have now clarified this in the manuscript (L173-176). 

L143: where is the data of soil moisture?

>> We do not have data on soil moisture.

L167: more details are needed on SI to introduce this index.

>> We have now incorporated more details about the SI index (L218-220).

L172: “total number of species”? I thought authors mostly identify the nematodes at genus level

>> The reviewer is correct, we identified most of our species to genus and morphospecies level. We have now replaced the word ‘species’ by ‘taxa’ throughout the manuscript.

L319-321: not understandable

>> We have deleted this sentence in the current version of our discussion.

L329: “have” should be “result in”

>> We have changed this as suggested by the reviewer.

L331: “both” should be “all three”

>> We have not changed this because the Enrichment Index (EI) was not significantly lower in invaded areas compared to managed areas, i.e. only SI and MI were significantly lower in invaded areas compared to managed areas. 

L334: “perturbations” should be “perturbations, such as plant invasions”

>> We have changed this as suggested by the reviewer.

L340: should be “soil nematode communities” instead of “communities”

>> We have incorporated the reviewer’s suggestion.

---

## [Decision Letter · Decision Letter 1]

12 Nov 2019

PONE-D-19-17858R1

Community- and trophic-level responses of soil nematodes to removal of a non-native tree at different stages of invasion

PLOS ONE

Dear Miss Peralta,

Thank you for submitting your manuscript to PLOS ONE. After careful consideration, we feel that it has merit but does not fully meet PLOS ONE’s publication criteria as it currently stands. Therefore, we invite you to submit a revised version of the manuscript that addresses the points raised during the review process.

We would appreciate receiving your revised manuscript by Dec 27 2019 11:59PM. To enhance the reproducibility of your results, we recommend that if applicable you deposit your laboratory protocols in protocols.io, where a protocol can be assigned its own identifier (DOI) such that it can be cited independently in the future. For instructions see: http://journals.plos.org/plosone/s/submission-guidelines#loc-laboratory-protocols

We look forward to receiving your revised manuscript.

Kind regards,

Martin Schädler

Academic Editor

PLOS ONE

Additional Editor Comments (if provided):

The manuscript is much improved. There are only few minor comments. However, I think some changes to the new Figure 4 are warranted prior to publication. In its current form, it is not immediately intuitive how the data are depicted.

Reviewers' comments:

Reviewer's Responses to Questions

**Comments to the Author**

1. If the authors have adequately addressed your comments raised in a previous round of review and you feel that this manuscript is now acceptable for publication, you may indicate that here to bypass the “Comments to the Author” section, enter your conflict of interest statement in the “Confidential to Editor” section, and submit your "Accept" recommendation.

Reviewer #1: (No Response)

Reviewer #2: All comments have been addressed

2. Is the manuscript technically sound, and do the data support the conclusions?

Reviewer #1: Yes

Reviewer #2: Yes

3. Has the statistical analysis been performed appropriately and rigorously? 

Reviewer #1: Yes

Reviewer #2: Yes

4. Have the authors made all data underlying the findings in their manuscript fully available?

Reviewer #1: Yes

Reviewer #2: Yes

5. Is the manuscript presented in an intelligible fashion and written in standard English?

Reviewer #1: Yes

Reviewer #2: Yes

6. Review Comments to the Author

Reviewer #1: I appreciate the efforts of the authors to revise this manuscript. The manuscript is much improved, particularly regarding the new analyses and restructuring of the Introduction and Discussion. The authors have highlighted the original Dickie et al. 2014 paper in AoB Plants, and how this manuscript builds upon that published work. Additionally, the authors provide greater methodological details and justification for the tree, sapling, and seedling removal management treatments. Specific plans for making the data publicly available have now been stated, complying with the PLOS data policy. I have only a few concerns to be addressed, outlined below.

Results:

S2 Table: Table caption isn’t overly informative. This table appears to summarize results comparing the seedling removal treatment to the other treatments, so should be stated in the table caption.

Figure 4: I’m a bit confused by this figure. For relative abundance data, figures are normally shown in percentages. Since an unequal number of nematodes were identified per sample ('approximately 100'), standardizing those to values to percentages would be most appropriate. A quick glance at Figure 4 reveals that adding up the bars of relative abundance for T1, T2, and T3 do not appear to sum to 100. That makes it harder to attribute differences to treatments versus sampling effort. It may be worthwhile to offer a more thorough explanation in the figure legend as well.

Minor comments:

Line 78: change ‘removal or…’ to ‘removal of…’

Line 329: I think you mean ‘were’ not ‘where’

Line 350: change ‘completed’ to ‘completely’

Reviewer #2: The authors have considered my and other reviewer comments and made satisfactory and considerable changes in the new version of the manuscript "Community- and trophic-level responses of soil nematodes to removal of a non-native tree at different stages of invasion". The text has been edited significantly and the readability and flow of the manuscript has greatly improved. The change of an informative title and the reanalysis of the data, the addition of associated visual elements have resulted in a more interesting manuscript. Overall, this manuscript has been improved significantly.

7. PLOS authors have the option to publish the peer review history of their article (what does this mean?). If published, this will include your full peer review and any attached files.

Reviewer #1: No

Reviewer #2: Yes: Minggang Wang

---

## [Author Response · Author response to Decision Letter 1]

20 Nov 2019

Response to reviewers

Additional Editor Comments (if provided):

The manuscript is much improved. There are only few minor comments. However, I think some changes to the new Figure 4 are warranted prior to publication. In its current form, it is not immediately intuitive how the data are depicted.

>> We appreciate the Editor and reviewers’ positive feedback and comments. We have incorporated all the comments made by the reviewers (including a modified version of Figure 4) and responded specifically to each comment below, marking our responses with ‘>>’.

Reviewers' comments:

Reviewer's Responses to Questions

Comments to the Author

1. If the authors have adequately addressed your comments raised in a previous round of review and you feel that this manuscript is now acceptable for publication, you may indicate that here to bypass the “Comments to the Author” section, enter your conflict of interest statement in the “Confidential to Editor” section, and submit your "Accept" recommendation.

Reviewer #1: (No Response)

Reviewer #2: All comments have been addressed

>> We have modified Figure 4 and incorporated all the minor comments suggested by Reviewer #1.

2. Is the manuscript technically sound, and do the data support the conclusions?

Reviewer #1: Yes

Reviewer #2: Yes

3. Has the statistical analysis been performed appropriately and rigorously?

Reviewer #1: Yes

Reviewer #2: Yes

4. Have the authors made all data underlying the findings in their manuscript fully available?

Reviewer #1: Yes

Reviewer #2: Yes

5. Is the manuscript presented in an intelligible fashion and written in standard English?

Reviewer #1: Yes

Reviewer #2: Yes

6. Review Comments to the Author

Reviewer #1: I appreciate the efforts of the authors to revise this manuscript. The manuscript is much improved, particularly regarding the new analyses and restructuring of the Introduction and Discussion. The authors have highlighted the original Dickie et al. 2014 paper in AoB Plants, and how this manuscript builds upon that published work. Additionally, the authors provide greater methodological details and justification for the tree, sapling, and seedling removal management treatments. Specific plans for making the data publicly available have now been stated, complying with the PLOS data policy. I have only a few concerns to be addressed, outlined below.

Results:

S2 Table: Table caption isn’t overly informative. This table appears to summarize results comparing the seedling removal treatment to the other treatments, so should be stated in the table caption.

>> We have changed the S2 Table caption as suggested by the reviewer. It now reads: 

“S2 Table. Results of generalised linear models with Poisson error distribution comparing taxa richness of the seedling removal management strategy with taxa richness of the sapling removal, no removal and tree removal management strategies. TL = trophic level. Bold values indicate significant results (α = 0.05).”

Figure 4: I’m a bit confused by this figure. For relative abundance data, figures are normally shown in percentages. Since an unequal number of nematodes were identified per sample ('approximately 100'), standardizing those to values to percentages would be most appropriate. A quick glance at Figure 4 reveals that adding up the bars of relative abundance for T1, T2, and T3 do not appear to sum to 100. That makes it harder to attribute differences to treatments versus sampling effort. It may be worthwhile to offer a more thorough explanation in the figure legend as well.

>> We appreciate the comment and apologise for the imprecision. We have now modified Figure 4 following the reviewer suggestion. Specifically, we replaced total abundances by percentages as suggested by the reviewer. We have also expanded the figure legend to clarify that the figure shows the percentage composition of each nematode taxon within a sampling site across management strategies and trophic levels.

Minor comments:

Line 78: change ‘removal or…’ to ‘removal of…’

>> We have changed this as suggested.

Line 329: I think you mean ‘were’ not ‘where’

>> We have changed this as suggested.

Line 350: change ‘completed’ to ‘completely’

>> We have changed this as suggested.

Reviewer #2: The authors have considered my and other reviewer comments and made satisfactory and considerable changes in the new version of the manuscript "Community- and trophic-level responses of soil nematodes to removal of a non-native tree at different stages of invasion". The text has been edited significantly and the readability and flow of the manuscript has greatly improved. The change of an informative title and the reanalysis of the data, the addition of associated visual elements have resulted in a more interesting manuscript. Overall, this manuscript has been improved significantly.

>> We appreciate the positive appraisal of our manuscript.

---

## [Editor Report · Decision Letter 2]

13 Dec 2019

Community- and trophic-level responses of soil nematodes to removal of a non-native tree at different stages of invasion

PONE-D-19-17858R2

Dear Dr. Peralta,

We are pleased to inform you that your manuscript has been judged scientifically suitable for publication and will be formally accepted for publication once it complies with all outstanding technical requirements.

With kind regards,

Martin Schädler

Academic Editor

PLOS ONE
---

## [Editor Report · Acceptance letter]

23 Dec 2019

PONE-D-19-17858R2 

Community- and trophic-level responses of soil nematodes to removal of a non-native tree at different stages of invasion 

Dear Dr. Peralta:

I am pleased to inform you that your manuscript has been deemed suitable for publication in PLOS ONE. Congratulations! Your manuscript is now with our production department. 

With kind regards,

on behalf of

Dr. Martin Schädler 

Academic Editor

PLOS ONE